# TRIPOD statement: a preliminary pre-post analysis of reporting and methods of prediction models

Amir H Zamanipoor Najafabadi ![ORCID],[1] Chava L Ramspek ![ORCID],[2] Friedo W Dekker,[2] Pauline Heus ![ORCID],[3] Lotty Hooft,[4] Karel G M Moons,[5] Wilco C Peul,[1,6] Gary S Collins,[7] Ewout W Steyerberg,[8] Merel van Diepen[2]

For numbered affiliations see end of article.

**Correspondence to**
Dr Amir H Zamanipoor Najafabadi;
a.h.zamanipoor_najafabadi@lumc.nl

## ABSTRACT

**Objectives** To assess the difference in completeness of reporting and methodological conduct of published prediction models before and after publication of the Transparent Reporting of a multivariable prediction model for Individual Prognosis Or Diagnosis (TRIPOD) statement.

**Methods** In the seven general medicine journals with the highest impact factor, we compared the completeness of the reporting and the quality of the methodology of prediction model studies published between 2012 and 2014 (pre-TRIPOD) with studies published between 2016 and 2017 (post-TRIPOD). For articles published in the post-TRIPOD period, we examined whether there was improved reporting for articles (1) citing the TRIPOD statement, and (2) published in journals that published the TRIPOD statement.

**Results** A total of 70 articles was included (pre-TRIPOD: 32, post-TRIPOD: 38). No improvement was seen for the overall percentage of reported items after the publication of the TRIPOD statement (pre-TRIPOD 74%, post-TRIPOD 76%, 95% CI of absolute difference: −4% to 7%). For the individual TRIPOD items, an improvement was seen for 16 (44%) items, while 3 (8%) items showed no improvement and 17 (47%) items showed a deterioration. Post-TRIPOD, there was no improved reporting for articles citing the TRIPOD statement, nor for articles published in journals that published the TRIPOD statement. The methodological quality improved in the post-TRIPOD period. More models were externally validated in the same article (absolute difference 8%, post-TRIPOD: 39%), used measures of calibration (21%, post-TRIPOD: 87%) and discrimination (9%, post-TRIPOD: 100%), and used multiple imputation for handling missing data (12%, post-TRIPOD: 50%).

**Conclusions** Since the publication of the TRIPOD statement, some reporting and methodological aspects have improved. Prediction models are still often poorly developed and validated and many aspects remain poorly reported, hindering optimal clinical application of these models. Long-term effects of the TRIPOD statement publication should be evaluated in future studies.

## Strengths and limitations of this study

► This is the first study to assess the completeness of reporting and methodological conduct of prediction models published before and after publication of the Transparent Reporting of a multivariable prediction model for Individual Prognosis Or Diagnosis (TRIPOD) statement.

► A limitation of this study is the short time period evaluated and therefore future studies are needed to assess the long-term effects on completeness of reporting and methodological conduct.

► Causality between publication of the TRIPOD statement and the found results cannot be established due to confounding.

## INTRODUCTION

Prediction models cover both prognostic models, which aim to predict the risk of future outcomes, and diagnostic models, which aim to assess the presence or absence of a condition.[1] They provide information for differential diagnosis, additional testing and for patient selection on treatment. Interest in prediction models has sharply increased over the last two decades, translating to new methodological developments, especially regarding performance assessment of these models.[2–4] In addition, clinical guidelines are increasingly recommending the use of prediction models,[5 6] and consequently implementation of these models in clinical practice for individualised diagnostic and therapeutic decisions has surged.[7–10]

Previous systematic reviews on the quality of published prediction models have identified poor reporting and many methodological shortcomings in the development and validation of these models.[11–13] In response to these reviews, the Transparent Reporting of a multivariable prediction model for Individual Prognosis Or Diagnosis (TRIPOD) statement was developed.[14] The TRIPOD statement provides reporting recommendations for articles that describe the development and external validation of prediction models, aiming to enhance reporting transparency and hence interpretability, reproducibility

and clinical usability of these models.[14] Although the TRIPOD statement primarily focuses on reporting and not on methods, current accepted methods for the development and validation of prediction models are discussed in the accompanying Explanation and Elaboration document.[15]

The primary aim of this study was to assess the difference in completeness of reporting and methodological conduct of published prediction models before and after publication in high impact general medicine journals.

## METHODS

### Systematic literature search

We selected the seven general medicine journals with the highest Web of Knowledge impact factor in 2017: *New England Journal of Medicine, Journal of the American Medical Association (JAMA), The Lancet, the British Medical Journal (The BMJ), Annals of Internal Medicine, PLOS Medicine, and BMC Medicine*. Articles on prediction models published in these journals before publication of the TRIPOD statement (pre-TRIPOD: 01 January 2012 to 31 December 2014) and after publication of the TRIPOD statement (post-TRIPOD: 01 January 2016 to 31 December 2017) were identified by a PubMed search string (online supplementary text 1). Articles published in 2015 were excluded from the search, as the TRIPOD statement was published in 2015 and we regard this as a transition period. Titles and abstracts were screened by one reviewer (AHZN). Full-text articles were screened by two independent reviewers (AHZN and CLR) and disagreement was resolved by discussion and consensus with a senior author (MvD).

### Article and model selection

Original articles with the primary aim of developing and/or validating multivariable models, both prognostic and diagnostic, were included. We excluded aetiological studies, genetic marker studies and model impact studies, as these are not covered by the TRIPOD statement. Included articles were classified as (1) Development, (2) Development and external validation, (3) External validation, and (4) Extension/updating of models. For articles addressing multiple models but not explicitly recommending a single model, the model with the most predictors was evaluated. For instance, Hippisley-Cox (2013) described model A, B and C for the prediction of future risk of cardiovascular disease, with model B being the same as model A with the addition of several predictors and interactions and model C being the same as model B with the addition of one variable. In this case model C was evaluated.

### Assessment of adherence to TRIPOD criteria

In 2018, authors of the TRIPOD statement published a TRIPOD adherence assessment form and adherence scoring rules, which were also used in our study.[16–18] The TRIPOD adherence form is a measurement tool developed for authors who want to evaluate the adherence of prediction model studies to TRIPOD, for example, over time or in a certain medical domain. In general, when multiple aspects were described within a TRIPOD item, all aspects needed to be reported to score a point for that item. For instance, the item title contains four subitems (eg, (1) Identifying the study as development and/or validation of a (2) prediction model with (3) description of target population and (4) outcome) and all four aspects need to be reported to score a point for this specific item. For all items and aspects of the checklist it was assessed whether it was reported in the main article or online supplementary materials. The main analyses were based on items reported in either the main text or supplements. Each article was only assessed for items applicable to the study (ie, development and/or external validation, or incremental value study). Scores for reporting level were calculated by assigning a single point for each reported item applicable to the study and total reporting level scores were converted to percentages based on the maximum possible score, and followed published scoring rules for the TRIPOD adherence form.[16 17]

### Assessment of study characteristics and used methods

In addition to the completeness of reporting following the TRIPOD statement, we assessed specific study characteristics and methods used in the included articles. To this end, we developed a comprehensive data extraction form based on previous studies, current methodological consensus, and the TRIPOD Exploration and Elaboration document (online supplementary text 1).[11 13 15 19–22] In summary the following topics were assessed: general study characteristics (ie, diagnostic or prognostic and study topic), handling of missing data, model development methods, type of external validation and updating, and performance measures. To facilitate interpretation of the results section, main recommendations of the TRIPOD Exploration and Elaboration document are presented in table 1. Assessment of these items was performed by two independent reviewers (AHZN and CLR) and a senior author (MvD) where necessary. In addition, for all articles published in the post-TRIPOD period, we extracted whether authors cited or referred to the TRIPOD statement, provided the completed checklist, if the article was published in a journal that published the statement (*The BMJ, Annals of Internal Medicine, BMC Medicine*) or was published in a journal that clearly stated in the author guidelines that they required TRIPOD adherence for submitted work at the time of writing this manuscript (*The BMJ, JAMA and PLOS Medicine*). While all included journals (except for the *NEJM*) encouraged authors to follow the Equator Network guidelines, which includes the TRIPOD checklist, in their author instructions, only *The BMJ, JAMA and PLOS Medicine* required adherence to the Equator network guidelines and also required to include a filled-out checklist at the time of submission.

**Table 1** Recommended methods and analyses for the development and validation of prediction models including supportive references

**Methodology**

| | | |
|---|---|---|
| Handling of missing data | It is generally advised to use multiple imputation for handling of missing data. Complete case analysis, single or mean imputation are inefficient methods to estimate coefficients. | 47–49 |
| Selection and retaining of predictors in multivariable models | Predictor selection and retaining is preferably based on clinical knowledge and previous literature, instead of significance levels in univariable or stepwise analysis. | 22 26 27 |
| Internal validation | It is advised to internally validate the model to assess optimism in performance and reduce overfitting. An efficient method is bootstrapping; split-sample validation should be avoided. | 25 26 |
| Calibration | It is advised to assess the calibration of a model at external validation. The preferred method is a calibration plot, with intercept and slope, and not statistical tests (eg, Hosmer-Lemeshow), as a plot retains the most information on possible miscalibration. | 22 26 27 50 |
| External validation | External validation of models is needed for rigorous assessment of performance. The preferred external validation population is fully independent. | 28 51 |

## Analysis and reporting of results

Reporting levels are presented as percentages, stratified by journal, and for comparison the absolute difference in percentages with 95% CIs are reported. Analyses were performed with IBM SPSS statistics (V.23.0, Armonk, New York, USA). Main results of the completeness of both reporting and methods are reported in text and detailed results are reported in the online supplementary tables. Comparisons were made between articles (1) pre-TRIPOD and post-TRIPOD, (2) post-TRIPOD between articles published in journals that published and did not publish the TRIPOD, (3) between articles published in journals that require TRIPOD adherence or not, (4) citing versus not citing the TRIPOD, and (5) providing versus not providing a completed TRIPOD checklist. Furthermore, to estimate changes over time regardless of the TRIPOD statement, a comparison was made between pre-TRIPOD articles and post-TRIPOD articles not citing the TRIPOD.

## Patient and public involvement

Patients and/or the public were not involved in the design, or conduct, or reporting, or dissemination plans of this research.

## RESULTS
## Characteristics of included studies

The PubMed search string retrieved 481 articles, of which the full texts of 119 were read and 70 met our inclusion criteria (pre-TRIPOD: 32 articles, post-TRIPOD: 38 articles, figure 1, online supplementary text 1). Most of the included articles were published in *The BMJ* (n=38), and least in *The Lancet* (n=3) and *NEJM* (n=1). In both the pre-TRIPOD and post-TRIPOD periods the majority of articles described prognostic models (as opposed to diagnostic models) and this increased in the post-TRIPOD period (pre-TRIPOD: 59%, post-TRIPOD: 89%)

(table 2). In the post-TRIPOD period the percentage of articles describing both the development and validation of a model (pre-TRIPOD: 31%, post-TRIPOD: 39%) or solely the external validation (pre-TRIPOD: 13%, post-TRIPOD: 26%) increased too. Thirty-two per cent of articles only described the development of a prediction model without external validation in the post-TRIPOD period, compared with 44% in the pre-TRIPOD period.

The majority of models were developed and/or validated using data from observational cohorts (pre-TRIPOD: 81%, post-TRIPOD: 82%) compared with other study designs such as randomised trials. More than half of the articles published in the post-TRIPOD period referred to the TRIPOD statement (n=20, 53%) and were published in journals that published the TRIPOD statement (n=21, 55%). The TRIPOD statement was cited in 48% of articles published in journals that published the TRIPOD, and in 59% of articles in journals that did not publish TRIPOD.

## Assessment of adherence to TRIPOD statement

Using the 2018 TRIPOD adherence assessment form, a minimal non-significant increase in the overall percentage of reported items was found comparing the pre-TRIPOD period (74%) with the post-TRIPOD period (76%, absolute difference 2%, 95% CI –4% to 7%, figure 2, online supplementary table 1), with no clear trend over the years (online supplementary figure 1). Results were similar for the comparison between pre-TRIPOD articles and post-TRIPOD articles not citing the statement (76%, absolute difference 2%, 95% CI –5% to 9%). An improvement for 16 of the individual TRIPOD items (44% of items, online supplementary table 2) was seen, while 3 (8%) of items showed no improvement and 17 (47%) items showed a decrease in the percentage of articles appropriately reporting the item. Post-TRIPOD, for articles referring versus not referring to the statement, published in

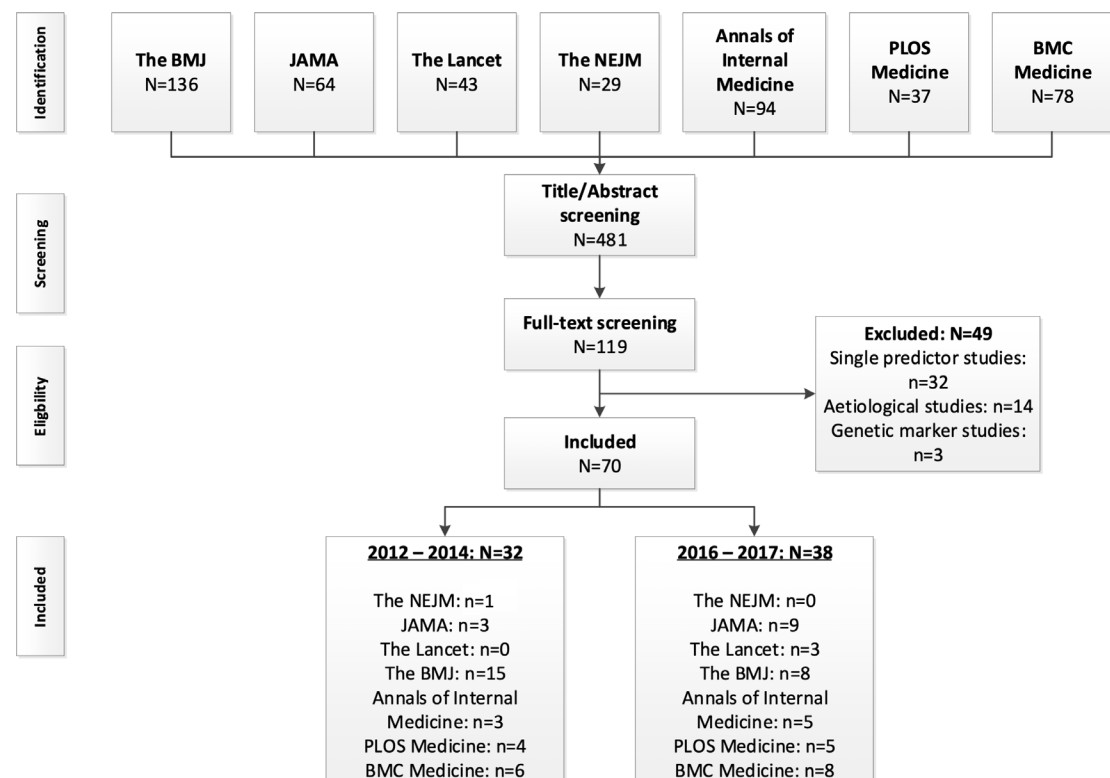

**Figure 1** Flow chart of search results and selection procedure. BMJ, British Medical Journal; JAMA, Journal of the American Medical Association; NEJM, New England Journal of Medicine.

journals that published versus did not publish the statement, and published in journals that required adherence to the statement versus did not require adherence to the statement, no difference in the completeness of reporting was observed (online supplementary tables 3-5). Five articles presented the completed TRIPOD checklist in the supplementary material and the overall percentage of reporting for these articles was 80%. The percentage of articles reporting TRIPOD items in their supplement is presented in online supplementary table 6.

### Assessment of specific TRIPOD items
#### Abstract
In both the pre-TRIPOD (16%) and post-TRIPOD period (8%), most abstracts did not report all the proposed subitems (TRIPOD item 2).

#### Reporting of missing data
In general, the reporting of missing data (TRIPOD item 13b) improved from 59% in the pre-TRIPOD period to 71% in the post-TRIPOD period, though fewer studies reported missingness per predictor in the post-TRIPOD period (pre-TRIPOD: 53%, post-TRIPOD: 37%, figure 3 and online supplementary table 7). Most studies did not report the reason for missing data (pre-TRIPOD: 84%, post-TRIPOD: 95%).

#### Model development and presentation
In the post-TRIPOD period, proper description of the characteristics of study participants (TRIPOD item 13b) was less often reported (37%) than in the pre-TRIPOD

period (50%). In the post-TRIPOD period, method of predictor selection (TRIPOD-item 10b) was more often reported (70%) than in the pre-TRIPOD period (62%), as was internal validation (TRIPOD-item 10b) of the developed model (pre-TRIPOD 62%, post-TRIPOD 74%). If performed, unadjusted analyses were less often reported (TRIPOD item 14b) in the post-TRIPOD period (64%) than in the pre-TRIPOD period (86%). In the post-TRIPOD period, the full model (ie, intercept or baseline hazard and all regression coefficients: TRIPOD-item 15a) was presented more frequently (42%), compared with the pre-TRIPOD period (27%). However, in both eras some studies still reported no information at all on the final model (pre-TRIPOD 8%; post-TRIPOD 4%, figure 3 and online supplementary table 8). To improve clinical usability (TRIPOD-item 15b), more than a third of studies reported to have developed a web application (pre-TRIPOD: 38%, post-TRIPOD: 37%) and some studies provided a simplified clinical risk score or nomogram (pre-TRIPOD: 29%; post-TRIPOD: 26%).

#### Performance measures
The percentage of studies reporting calibration (TRIPOD-item 16) of the model increased from 66% in the pre-TRIPOD period to 87% in the post-TRIPOD period. Discrimination (TRIPOD-item 16), was reported by all studies in the post-TRIPOD period and by 91% of studies in the pre-TRIPOD period. Measures of classification were reported less frequently in the post-TRIPOD period (pre-TRIPOD: 69%, post-TRIPOD: 58%). Measures of clinical

| Table 2 | Characteristics of included studies | |
| --- | --- | --- |
| | Before 2015 (n=32) number, (%) | After 2015 (n=38) number, (%) |
| Diagnostic/prognostic | | |
| Diagnostic | 13 (41%) | 4 (11%) |
| Prognostic | 19 (59%) | 34 (89%) |
| Type | | |
| Development | 14 (44%) | 12 (32%) |
| Validation | 4 (13%) | 10 (26%) |
| Development and validation | 10 (31%) | 15 (39%) |
| Update | 4 (13%) | 1 (3%) |
| Setting | | |
| General population and primary care | 18 (56%) | 18 (47%) |
| Secondary care | 14 (44%) | 20 (53%) |
| Design | | |
| Cohort | 26 (81%) | 31 (82%) |
| RCT | 1 (30%) | 4 (11%) |
| Cohort and RCT | 2 (6%) | 3 (8%) |
| Case-control | 3 (9%) | 0 (0%) |
| Topic | | |
| (Cardio)vascular | 12 (38%) | 16 (42%) |
| Oncological | 3 (9%) | 8 (21%) |
| Other | 17 (53%) | 14 (37%) |

RCT, randomised controlled trial.

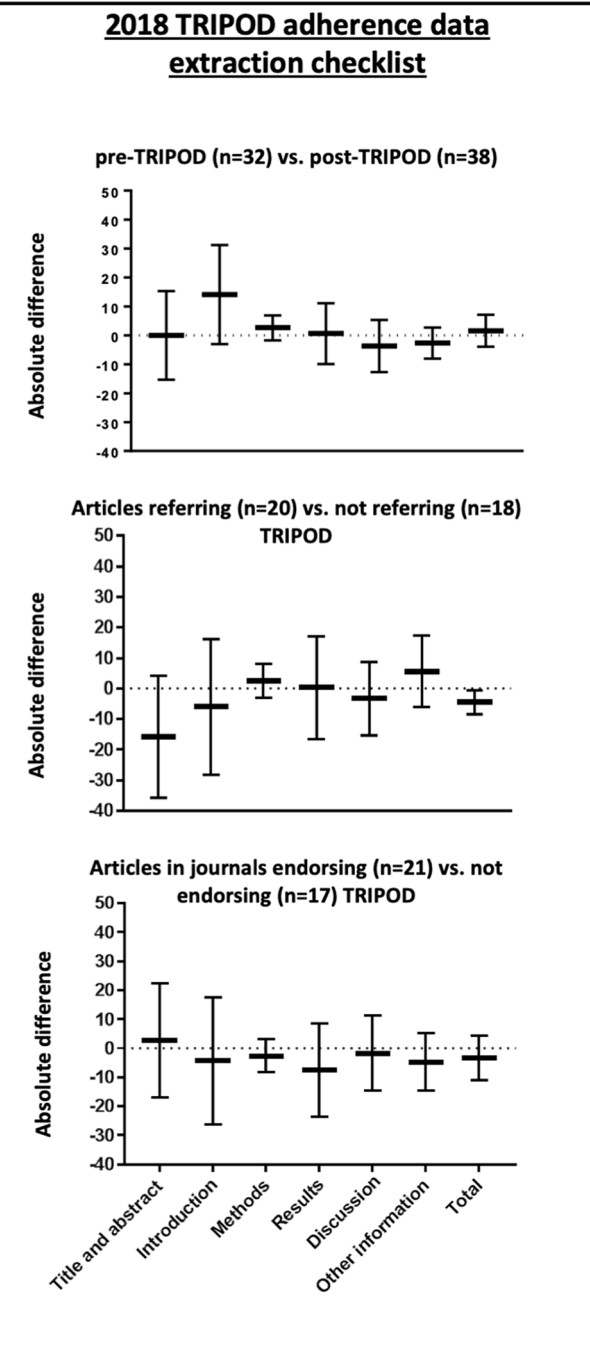

**Figure 2** TRIPOD reporting scores. TRIPOD, Transparent Reporting of a multivariable prediction model for Individual Prognosis Or Diagnosis.

usefulness like decision curve analysis were only reported by 2 (6%) studies in the pre-TRIPOD period and 7 (21%) studies in the post-TRIPOD period. Measures of overall performance like the Brier Score or $R^2$ were infrequently reported in both periods (pre-TRIPOD: 19%, post-TRIPOD: 21%). Detailed results are depicted in online supplementary table 7.

### Assessment of methods
#### Handling of missing data
Multiple imputation was the most frequently performed approach for handling missing data (pre-TRIPOD: 38%, post-TRIPOD: 50%). The number of studies that used a complete case analysis remained constant and was 16% in both the pre-TRIPOD and post-TRIPOD periods.

#### Model development
Post-TRIPOD, the number of studies that included predictors based on significance levels in univariable analysis decreased (pre-TRIPOD: 67%, post-TRIPOD: 44%, figure 2 and online supplementary table 8) as well as the number of studies using stepwise methods to retain predictors (pre-TRIPOD: 63%, post-TRIPOD: 48%). In general, a larger number of candidate predictors was used in the post-TRIPOD period (median: 25), compared

with the pre-TRIPOD period (median: 20). Internal validation was more frequently performed in the post-TRIPOD period (74%) compared with the pre-TRIPOD period (62%). When internal validation was performed, bootstrapping was the most frequently used method in both time periods with an increase from 29% in the pre-TRIPOD period to 41% in the post-TRIPOD period.

#### Performance measures
The majority of studies presented measures of calibration (pre-TRIPOD: 66%, post-TRIPOD: 87%) and discrimination

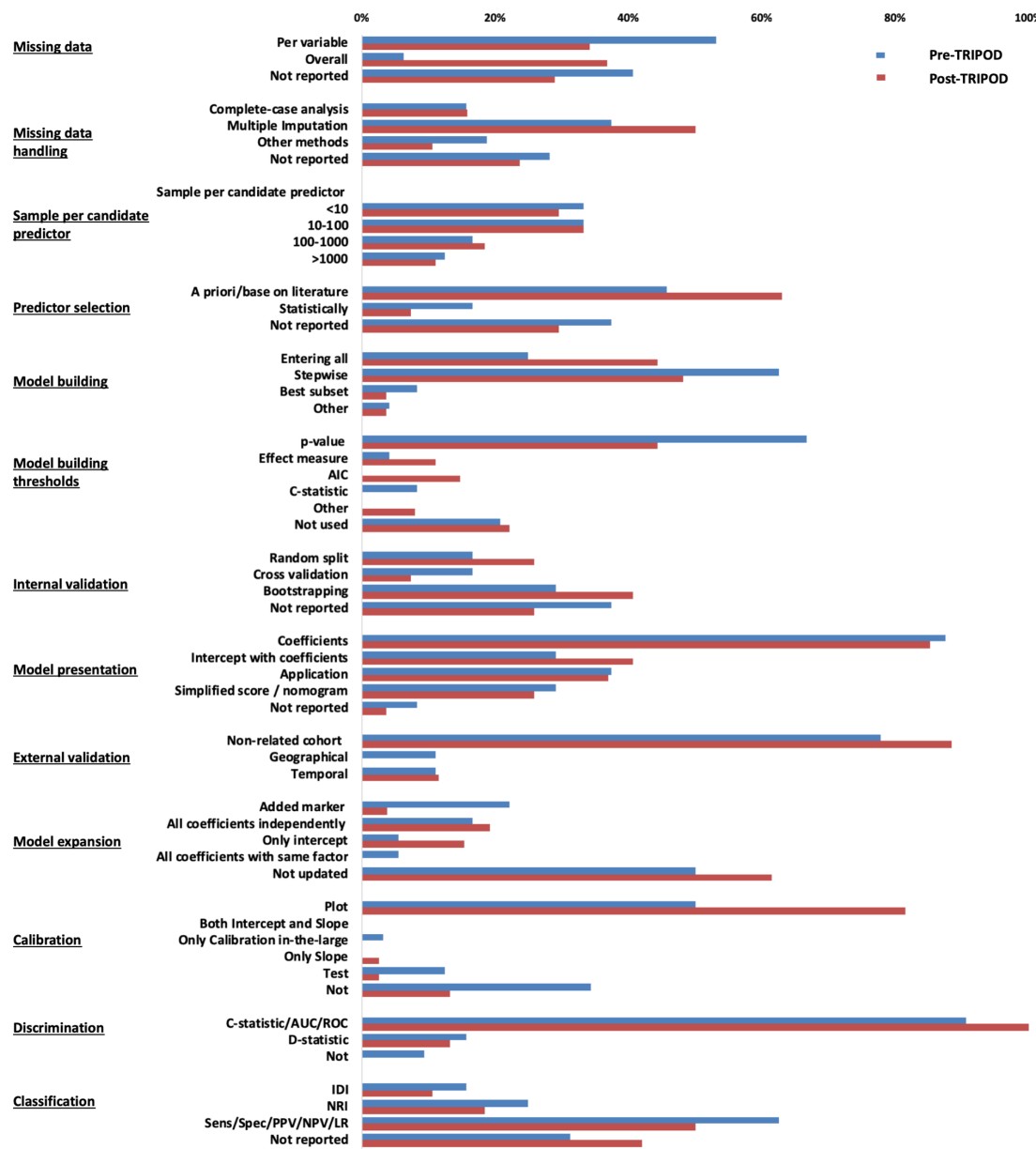

**Figure 3** Comparison of used methods in the pre-TRIPOD and post-TRIPOD period. AIC, Akaike Information Criterion; AUC, Area Under the Curve; IDI, Integrated Discrimination Improvement; LR, Likelihood Ratio; NPV, Negative Predictive Value; NRI, Net Reclassification Improvement; PPV, Positive Predictive Value; ROC, Receiver Operating Characteristics; Sens, Sensitivity; Spec, Specificity; TRIPOD, Transparent Reporting of a multivariable prediction modelfor Individual Prognosis Or Diagnosis.

(pre-TRIPOD: 91%, post-TRIPOD: 100%, figure 3 and online supplementary table 7). Calibration was primarily assessed with a calibration plot and this increased in the post-TRIPOD period (pre-TRIPOD: 50%, post-TRIPOD: 82%). Discrimination was primarily assessed with the C-statistic and area under the curve methods (pre-TRIPOD: 91%, post-TRIPOD: 100%). Measures of classification were reported in more than half of the studies (pre-TRIPOD: 69%, post-TRIPOD: 58%), mostly assessed with diagnostic test summary statistics (ie, sensitivity, specificity and positive and negative predictive values) (pre-TRIPOD: 63%, post-TRIPOD: 50%), and to a lesser extent the integrated discrimination improvement (pre-TRIPOD: 16%, post-TIRPOD: 11%) or the net reclassification improvement (pre-TRIPOD 25%, post-TRIPOD: 18%).

### External validation and model updating
Most external validation studies performed the validation in individuals fully unrelated to the development cohort (pre-TRIPOD 78%, post-TRIPOD: 88%, figure 3 and online supplementary table 9). Models were updated with an additional predictor in four (22%) studies before the TRIPOD statement and in one (4%) study after the TRIPOD statement.

### DISCUSSION
No significant improvement in the overall reporting quality of prediction models published in the seven general medicine journals with the highest impact factor was found in the post-TRIPOD period, according to the

TRIPOD adherence form. However, an improvement in general methodological conduct was found. Notably, more studies described external validation of a model, reported information on missing data, used multiple imputation methods instead of complete case analysis for handling of missing data, selected and maintained variables in multivariable models based on clinical relevance instead of statistical cut-offs, and assessed both discrimination and calibration measures. While improvement was found for almost half of the TRIPOD items, no improvement or a deterioration was found for the other half of the items.

## Recommendations on reporting and methods

Though improvements over time in specific aspects of reporting and methods were apparent, there is room for further progress. While an increase in studies reporting the percentage of missing data in the post-TRIPOD period was observed, the amount of missingness was often not reported per predictor, yet this is important for the assessment of clinical usability of the model.[15] Multiple imputation was the most frequently performed method for handling missing data, which generally is the preferred approach.[23] Reporting of all coefficients of the final multivariable model and intercept, which is necessary for external validation and clinical use of models, increased over time.[22] Although widely discouraged, a number of studies in both the pre-TRIPOD and post-TRIPOD periods included predictors in multivariable prediction models based on data-driven selection methods such as univariable significance and/or stepwise methods. Such methods increase the risk of overfitted and poorly calibrated models.[11 22–26] Instead, it is advised to select predictors based on clinical knowledge and previous literature.[27] While the percentage of studies that both developed and externally validated a model increased over time, still more than 30% of articles only described the development of a model. External validation in a fully independent cohort is strongly recommended, as model performance might significantly decrease in cohorts other than the development cohort.[28] Assessment of both calibration and discrimination also increased, which is necessary in order to judge a model's predictive accuracy. Calibration refers to the agreement between absolute predicted and observed outcomes and the majority of studies used the preferred calibration plot.[29] Discrimination, a relative measure on the ability to distinguish between patients with and without the outcome, was reported by almost all studies.[29]

## Comparison with other reporting guidelines

A large number of reporting guidelines have been published for various study types.[19 30–33] Mixed results on the effect of these guidelines on the completeness of reporting have been found.[34–38] While an overall modest improvement in reporting was described for randomised controlled trials after publication of the CONSORT (Consolidated Standards of Reporting Trials) statement and by the STARD (Standards for Reporting Diagnostic accuracy studies) statement for diagnostic studies, no clear improvement was described for observational studies by the STROBE (Strengthening The Reporting of OBservational Studies in Epidemiology) statement and prognostic marker studies by the REMARK (Reporting Recommendations for Tumor Marker Prognostic Studies) guideline as described by the authors of these studies.[34–38] These findings pose the question how the introduction and publication of these guidelines can optimally impact the research field. For both the CONSORT and STARD statements, journals endorsing the statement showed a higher level of reporting compared with journals not endorsing these statements. Nevertheless, this was not found for the REMARK guideline, nor in our study for the TRIPOD statement.[34 37 38] Evidence of a relation between citing the statement and reporting level is also limited, as no association between this was found for the STARD nor in our study.[38] Requiring authors to provide and publish the completed checklist might help to improve reporting levels, as we found that the small numbers of studies providing the checklist reported more items on average. Therefore, we do recommend journals to ask authors to submit the completed checklist on submission, and require authors to publish it as a supplement, and reviewers and editors to control the provided checklist. However, as endorsing, citing and providing the checklist seems to have only a small effect on the reporting quality, we believe it is even more important to train methodologists and clinicians to interpret and use the checklist. This is supported by the results that even studies that provided the completed checklist, still did not report all items of the TRIPOD statement in our analysis of reporting. Documents such as the TRIPOD Exploration and Elaboration document facilitate proper interpretation, but we believe that the threshold to use this detailed document might be too high for the unexperienced researcher. Other possibilities to familiarise authors with the checklist should be explored, such as collaborative efforts of educational institutions and the TRIPOD committee to train researchers and clinicians. Online training courses might be of added value to reach a large target group.

## Comparison with other reviews on the completeness of reporting and methodological conduct of prediction models

Previous studies, published between 2012 and 2014, concluded poor reporting and use of methods for prediction models.[11 13 20 21] Comparing our results with a study assessing reporting and methods of prediction studies published in six high impact general medicine journals in 2008, improvement since then is clear for both methods and reporting. Considering methods more studies are externally validated, compose calibration plots to assess calibration and use multiple imputation for handling missing data. Improvement in reporting is also apparent as more studies report calibration and discrimination measures. Furthermore, a recently published article assessed the reporting quality of prediction models

published in 37 clinical domains in 2014 using the 2018 TRIPOD adherence assessment form, which found similar results to our pre-TRIPOD results.[17] As we only included articles published in high-impact general medicine journals it is difficult to generalise these results to the entire medical academic research field. We could argue that the improvement we observed might be an overestimation if general medical journals adopted the TRIPOD guidelines and new methodological insights with more speed and rigour. However, the opposite might also be true as these high-impact general medicine journals already had high methodological standards before the TRIPOD statement publication.[11 13 16 21 35 36 39 40]

### Strengths and limitations

A limitation of the current study is that the evaluation of studies was limited to the first 2 years after the TRIPOD statement publication. It may take some years before a reporting guideline is widely disseminated and accepted and the full impact is measurable. However, to somewhat overcome this problem we did not include any articles published in 2015, as the TRIPOD statement was published in January 2015 and we therefore saw this as a transition period. In addition, a previously published study on the effect of STARD found significant improvement within 2 years after publication.[38] Furthermore it is not possible to causally attribute the reported changes to the TRIPOD statement, as the results might be confounded by other developments in the last decade, such as publication of multiple series on the conduct of prediction models, publication of other guidelines such as the REMARK guideline for tumour marker prognostic studies and a general increase in the numbers of published prediction models.[41–43] One may also expect that authors who work in the field of prediction models are aware of the publication of the TRIPOD statement, especially those who publish in high-impact general medicine journals. A strength of the study is that the actual used methods for the development, description, validation and updating of prediction models were also assessed. While reporting and used methods are inherently related, the focus is different. A poorly developed model may be described fully and transparently in a manuscript and score high on reporting quality and vice versa a well-developed model may have poor reporting.[16] Furthermore, we have facilitated comparison to future TRIPOD reviews by using the 2018 TRIPOD adherence assessment form. Although both reporting and methods were comprehensively assessed, we might have missed interesting items for evaluation, especially as the field of prediction models is continuously developing. We also did not assess the risk of bias of the included studies with the PROBAST risk of bias assessment tool, as it would be not feasible to score the included articles according to the PROBAST, since to do so subject-specific knowledge is required and the included studies span a wide range of clinical subjects. Furthermore, as the PROBAST only gives suggestions for signalling questions and no scoring rules, it does

not completely fit with the aim to assess the actual used methods of the included studies. Furthermore, it would have been of interest to compare articles published in journals that between January 2016 and December 2017 obligated authors to complete the TRIPOD checklist; however this information was not available.

### Unanswered questions and future research

Future studies should focus on the long-term effects of the TRIPOD statement publication on reporting quality and methods, using the 2018 TRIPOD adherence form to allow for comparisons over time using the same adherence assessment tool. In addition, effects of the statement should be assessed in different medical fields for which a pre-TRIPOD baseline measurement is already performed.[16] Earlier studies on the effect of other reporting guidelines showed that the effect of these guidelines may be smaller or larger in specific medical fields.[34 40]

A new emerging field is the development of prediction models using artificial intelligence, machine learning and deep learning methods. In addition, more often omics data are used as predictors for these models.[44] While these models have many similarities with traditional regression methods, they differ in some aspects and may require specific guidelines on reporting.[44 45] Accordingly, the TRIPOD-Artificial Intelligence (AI) tool has recently been announced and is underway.[46] Similarly, reporting guidelines for prediction model impact studies are missing.

With the increasing number of reporting guidelines and lack of clear evidence that all guidelines improve reporting quality, research should be conducted to find methods to optimise the form, use and impact of these guidelines. With this in mind, there should also be focus on the overlap between different reporting guidelines. Prediction models can be reported following the TRIPOD statement, the STARD statement for diagnostic test accuracy studies and REMARK for prognostic tumour marker studies. As an increasing amount of studies contains multiple goals, analyses and data sources, it may be difficult to adhere to all applicable and relevant guidelines within the maximum word count. This holds especially for the abstract section of articles.

### CONCLUSION

No improvement was found comparing the post-TRIPOD period with the pre-TRIPOD period in the overall reporting quality of prediction models published in the seven general medicine journals with the highest impact factor. Comparison of articles published before the TRIPOD statement with non-TRIPOD citing articles published after the TRIPOD statement yielded similar results as the main pre-post comparison, further suggesting a lack of direct impact of the TRIPOD statement on overall reporting levels. However improvement was found in various specific aspects of methodological

conduct. More studies described external model validations, reported information on missing data, used multiple imputation methods for handling of missing data, reported the full prediction model and reported information on performance measures. However, there is still room for improvement in both the reporting and used methods of these models, as prediction models are still erroneously developed and validated and many aspects remain poorly reported, hindering optimal use of these models in clinical decision making. Long-term effects of the TRIPOD statement publication should be evaluated in future studies, ideally using the same 2018 TRIPOD adherence assessment form to allow for comparisons over time.

**Author affiliations**
[1]Department of Neurosurgery, Leiden University Medical Center, Leiden, The Netherlands
[2]Department of Clinical Epidemiology, Leiden University Medical Center, Leiden, The Netherlands
[3]Cochrane Netherlands, Julius Center for Health Sciences and Primary Care, University Medical Center (UMC) Utrecht, Utrecht University, Utrecht, The Netherlands
[4]Dutch Cochrane Centre (DCC), Julius Center for Health Sciences and Primary Care, University Medical Centre (UMC) Utrecht, Utrecht University, Utrecht, The Netherlands
[5]Department of Epidemiology, Julius Center for Health Sciences and Primary Care, Utrecht, The Netherlands
[6]Department of Neurosurgery, The Hague Medical Center, The Hague, The Netherlands
[7]NDORMS, Oxford University, Oxford, UK
[8]Department of Medical Statistics, Leiden University Medical Center, Leiden, The Netherlands

**Contributors** AHZN conceived the study. AHZN, MvD, CLR and EWS developed the study design with input from FWD, PH, LH, KGMM, WCP, GSC. AHZN and CLR screened the literature and performed the data extraction. AHZN performed the statistical analysis and wrote the first and successive drafts of the manuscript. AHZN, CLR, FWD, PH, LH, KGMM, WCP, GSC, EWS, MvD interpreted the data, critically revised the manuscript for important intellectual content, and approved the final version of the manuscript.

**Funding** AHZN and CLR were supported by personal Leiden University Medical Centre MD/PhD Scholarships. MvD was supported by a grant from the Dutch Kidney Foundation (16OKG12). GSC is supported by the NIHR Biomedical Research Centre, Oxford and Cancer Research UK programme grant (C49297/A29084).

**Competing interests** GSC, KGMM and EWS are members of the TRIPOD group. All authors have completed the ICMJE uniform disclosure form at http://www.icmje.org/coi_disclosure.pdf (available on request from the corresponding author) and declare no other support from any organisation for the submitted work than the grants reported in the funding section; no financial relationships with any organisations that might have an interest in the submitted work in the previous 3 years, no other relationships or activities that could appear to have influenced the submitted work.

**Patient consent for publication** Not required.

**Provenance and peer review** Not commissioned; externally peer reviewed.

**Data availability statement** All data relevant to the study are included in the article or uploaded as supplementary information. All data sets that were used are retrievable following the instruction of the original papers.

**ORCID iDs**
Amir H Zamanipoor Najafabadi http://orcid.org/0000-0003-2400-2070
Chava L Ramspek http://orcid.org/0000-0002-7883-5927
Pauline Heus http://orcid.org/0000-0002-6886-4652

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
