## [Reviewer comments · BMJ Open]

This paper was submitted to a another journal from BMJ but declined for publication following peer review. The authors addressed the reviewers' comments and submitted the revised paper to BMJ Open. The paper was subsequently accepted for publication at BMJ Open.

ARTICLE DETAILS

TITLE (PROVISIONAL)	TRIPOD statement: a preliminary pre-post analysis of reporting and methods of prediction models
AUTHORS	Zamanipour Najafabadi, Amir H.; Ramspek, Chava; Dekker, Friedo; Heus, Pauline; Hooft, Lotty; Moons, Karel; Peul, Wilco; Collins, Gary; Steyerberg, Ewout; van Diepen, Merel

VERSION 1 – REVIEW

REVIEWER	Qi Feng The University of Hong Kong
REVIEW RETURNED	30-Mar-2020

GENERAL COMMENTS	TRIPOD is an important guideline for comprehensive and throughout reporting of prediction model studies. In this interesting systematic review, the authors explored the impact of TRIPOD statement on reporting and methodological quality of prediction model studies, by making comparisons on characteristics of studies published on 7 journals before and after the publication of TRIPOD statement. Overall, the authors found no improvement in reporting completeness between pre- and post-TRIPOD periods, but some specific items in reporting and methodological quality have improved. They also found no difference between TRIPOD-citing studies and their counterparts, and between studies published on TRIPOD-endorsing journals and those published on non-TRIPOD-endorsing journals. The results imply that there is still room for improvement. However, I have some concerns. First, the number of included studies is small, with only 70 eligible prediction model studies in total, and 32 and 38 for pre- and post-TRIPOD periods, respectively, which reduces the accuracy of all estimates and the power for all comparisons. Previous reviews investigating the impact of published guidelines/statements generally included more studies than 70, for example, 211 studies for CONSORT (JAMA 2001;285(15):1992-5), 456 for STROBE (PLoS One 2013;8(8):e64733), 240 for STARD (Radiol 2008;248(3):817-23), 300 for CONSORT extension to clustered trials (BMJ 2011;343:d5886), 100 for PRISMA (Syst Rev 2017;6(1):263), etc. So, I would suggest the authors to extend the time frame and/or include more journals for literature search to gain a larger body of studies. Second, the current observational design
--

	and data analysis are insufficient to answer the research question stated in the last paragraph of the Introduction part "...to assess the impact of the TRIPOD statement on both...". Even if the authors had observed an improvement, it would be difficult to attribute it to the publication/dissemination of TRIPOD statement, due to potential confounding issues, such as increasing education or awareness. I would suggest that the authors also directly compare pre-TRIPOD studies and non-TRIPOD-citing studies, or studies published on non-TRIPOD-endorsing journals. This paper (PLoS One 2013;8(8):e64733) may also provide a view to address this issue. Third, I think the authors need to make a clearer definition on TRIPOD-endorsing journals. I guess the current definition for "endorsing" is a journal that published the original TRIPOD statement in 2015. However, it is possible that a journal did not publish TRIPOD statement but requires all manuscript submission adhering to TRIPOD (usually stated in its author instructions); vice versa. It would be better to differentiate the difference between them. I raised this concern because this study found 18 studies cited TRIPOD but 21 were published on TRIPOD-endorsing journals. This discrepancy looks interesting and may reflect some fundamental issues in exposure definition. My other minor concerns are listed below.  1. It seems that the "Recommendations on reporting and methods" in the Discussion provides nothing novel, and I would suggest making this part as concise as possible. 2. The authors may need to state in the Discussion that there may be other items for evaluation missed in this study, which is one of the limitations. 3. Page 4 line 105-107, "for articles addressing multiple models but not explicitly recommending a single model, the model with the most predictors was included". Do these models use the same outcome? Does that mean you would include the full model? It would be better to show an example. 4. Please unify "TRIPOD adherence form" and "TRIPOD Adherence form" through the manuscript. 5. Figure 1, bottom left box "The NEJM: n=n=1" should be "The NEJM: n=1". 6. The study assessed whether an item was reported in the main text or supplementary materials, and supplementary table 5 showed items reported in the supplementary materials. I wonder the reporting score represented in figure 2 was based on reporting in the main text only or in main text and supplementary materials both. 7. The authors may consider reporting in Results the percentage of studies that reported any discrimination measures, any classification measure, and any calibration measure.
--	--

REVIEWER	Junfeng Wang Julius Center for Health Sciences and Primary Care, UMC Utrecht
REVIEW RETURNED	03-Apr-2020

GENERAL COMMENTS	Zamanipoor Najafabadi et al. compared the adherence to TRIPOD statement before the publication of TRIPOD (pre-TRIPOD) and one year after that (post-TRIPOD). In total 70 articles were assessed (32 pre-TRIPOD vs 38 post-TRIPOD), and no significant improvement in percentage of reported items was found. From my experience in evaluating reporting quality of prediction models[1] and diagnostic tests[2], this very informative paper represents enormous amount of work done by the authors. So I
---

would like to congratulate them on finalizing all data extraction, statistical analysis and this high quality manuscript.

I listed my questions below: some are comments and some are for the curiosity of the reviewer and potential audiences.

Abstract

Line 42, page 2, "Quality of methods was assessed with a comprehensive data extraction form based on 43 previous studies, current methodological consensus, and the TRIPOD Exploration and Elaboration document." TRIPOD mainly focused on reporting completeness, although the E&E did include some methodology considerations, it is not the best tool for methodology quality assessment. PROBAST risk of bias assessment tool would be a better choice for this purpose.

Line 53, page 2, "More models were externally validated...." which is not clear enough. Models can be externally validated in other papers as well, I assume this is not the authors meant to say. Maybe better to use "More models were externally validated in the same paper...." here.

Introduction

Line 74, page 3, "Previous systematic reviews on published prediction models have identified poor reporting and many methodological shortcomings in the development and validation of these models." Systematic reviews of prediction models usually refer to reviews of model performance rather than methodological quality. This sentence should be more specific as "Previous systematic reviews on quality of published prediction models...".

Methods

Line 95, page 4, "PubMed search string (Supplementary text 1)". The reviewer checked the search strategy provided in Supplementary text 1, and suggests to add "nomogram" into the terms. Sometimes, nomogram is used to refer to a prediction model. Although the reviewer does not support this, a lot of researchers still like to use nomogram as the title of a clinical prediction model study, instead of using prediction, prognostic, or diagnostic. The authors may miss some prediction model studies if "nomogram" was not added to the search strategy.

Line 133, page 5, "In addition, we extracted whether authors cited or referred to the TRIPOD Statement...". This is a good point, it will be more helpful if information of whether TRIPOD checklist was provided in supplementary material was extracted as well. Implementation of TRIPOD contains several levels: (1) availability/awareness of TRIPOD (pre- vs post-); (2) use of TRIPOD (cited or referred); (3) correct understand of TRIPOD (providing a properly filled TRIPOD checklist. (1) is already achieved by the publication of TRIPOD in 2015, (2) will need promotion of TRIPOD to researchers, and (3) will need more training or workshop provided to researchers. So it is important to check for those authors filled out the checklist (usually this means all items were reported), did they indeed reported completely adherent to TRIPOD, so we can know what actions are needed to improve the reporting quality.

Line 139, page 5, "Reporting levels are presented as percentages...". The authors also presented the SD together with percentages, please specify the methods for calculating SD.

Because the reviewer tried to replicate the calculation of SD by using standard formula but did not get the same number as presented in Supplementary Table 1. And actually, the reviewer did not think it is necessary to add SD in that table.

Results

Line 166, page 6 “However, an improvement for almost half of the individual TRIPOD items (16 items, 42% of items, Supplementary Table 2) “. This sentence is misleading. When the reviewer read it for the first time, it seems a great improvement, but when read it again and looked at Supplementary Table 2, 58% of items were less frequently reported post-TRIPOD. Can we conclude this is an improvement? So the message is too positive, and should be balanced.

Line 193, page 7, “... almost half of the studies reported to have developed a web application (pre-TRIPOD: 38%, post-TRIPOD: 37%)...”. 38% and 37% are more near to one third than “almost half”. “Almost half” sounds like above 45%...

Discussion

Line 240, page 9, “a clear improvement in reporting and methodological conduct was found for certain aspects.” As the reviewer already mentioned in Results section, the improvement observed in 16 items is more like by chance. If reporting quality can either increase or decrease, we can always observe improvement in some of the items. So “a clear improvement ... for certain aspects” is just like cherry-picking.

It will be great if the authors can extensively discuss why reporting quality is still sub-optimal even after the publish of TRIPOD, and what efforts are needed to improve that. That will add value to the current descriptive report.

Tables and Figures

Page 18, Table 1. The heading of Table 1 should be more informative. For example: Analysis, Recommended methods, References.

Page 19, Table 2. The heading of Table 2 should be consistent with the data in the table: “number (%)”.

Page 20, Figure 1. Bottom left of the flow chart, “The NEJM: n=n=1” should be “The NEJM: n=1”

Page 21, Figure 2. “Ransom split” should be “Random split”; “Fully independent” is not clear, usually “domain validation” is used together with temporal and geographic validations; “Intercept and slope, calibration in-the-lagre, slope”, the first one includes the latter two, do the author mean “Both intercept and slope, only calibration in-the-lagre, only slope”? Why ROC is considered as classification but not part of C-statistic/AUC?

Supplementary documents

Page 24, Supplementary Table 1. As the reviewer mentioned in Methods section, SD can not be replicated. This also applies to Table 3 and Table 4.

Page 24, Supplementary Table 2. SD is in the heading, but not reported in the table.

	Page 27, Supplementary Table 6. The heading should be consistent with the data in the table: “number (%)”. Page 28, Supplementary Table 7. The heading should be consistent with the data in the table: “number (%)”. Page 29, Supplementary Table 8. The heading should be consistent with the data in the table: “number (%)”. [1] Liu, S., Chen, W., Heus, P., Dong, W., Zhai, W., Cui, Z., & Wang, J. (2019, July). Assessment of reporting quality of prediction model studies in HSCT: Adherence to the tripod statement. In BONE MARROW TRANSPLANTATION (Vol. 54, pp. 685-685). MACMILLAN BUILDING, 4 CRINAN ST, LONDON N1 9XW, ENGLAND: NATURE PUBLISHING GROUP. [2] Korevaar, D. A., Wang, J., van Enst, W. A., Leeflang, M. M., Hooft, L., Smidt, N., & Bossuyt, P. M. (2015). Reporting diagnostic accuracy studies: some improvements after 10 years of STARD. Radiology, 274(3), 781-789.
--	--

VERSION 1 – AUTHOR RESPONSE

Reviewer: 1 (Qi Feng)

Comments:

TRIPOD is an important guideline for comprehensive and throughout reporting of prediction model studies. In this interesting systematic review, the authors explored the impact of TRIPOD statement on reporting and methodological quality of prediction model studies, by making comparisons on characteristics of studies published on 7 journals before and after the publication of TRIPOD statement. Overall, the authors found no improvement in reporting completeness between pre- and post-TRIPOD periods, but some specific items in reporting and methodological quality have improved. They also found no difference between TRIPOD-citing studies and their counterparts, and between studies published on TRIPOD-endorsing journals and those published on non-TRIPOD-endorsing journals. The results imply that there is still room for improvement.

However, I have some concerns. First, the number of included studies is small, with only 70 eligible prediction model studies in total, and 32 and 38 for pre- and post-TRIPOD periods, respectively, which reduces the accuracy of all estimates and the power for all comparisons. Previous reviews investigating the impact of published guidelines/statements generally included more studies than 70, for example, 211 studies for CONSORT (JAMA 2001;285(15):1992-5), 456 for STROBE (PLoS One 2013;8(8):e64733), 240 for STARD (Radiol 2008;248(3):817-23), 300 for CONSORT extension to clustered trials (BMJ 2011;343:d5886), 100 for PRISMA (Syst Rev 2017;6(1):263), etc. So, I would suggest the authors to extend the time frame and/or include more journals for literature search to gain a larger body of studies.

We agree with the reviewer that the number of included studies is relatively small, and hence we might miss the power to detect a small significant improvement in overall reporting. However, by extending the number of general medicine journals we will include also journals with a very low impact factor (smaller than 6), while the aim of the study was to describe the impact of the TRIPOD statement in high impact general medicine journals, which is in line with previous reports of the impact of reporting guidelines. As these journals keep high rigorous standards, it is expected to see the

results of TRIPOD implementation in these journals first, while it can take many years before the effect is noticeable in topic-specific journals.

We examined the increase in the number of studies by extending the timeframe, which only resulted in a small increase in identified articles. This still doesn't allow for a large comparison similar to the reviews mentioned by the reviewer. Acknowledging these limitations, we would like to change the scope of the paper to "*evaluate the preliminary difference in completeness of reporting and methodological conduct of published prediction models before and after publication of the TRIPOD statement*". We have made the following changes:

- Title: "~~The impact of the~~ TRIPOD statement: a preliminary pre-post analysis of reporting and methods of prediction models"
- Abstract (page 2, lines 31-33): "To assess the difference in completeness of reporting and methodological conduct of published prediction models before and after publication of the Transparent Reporting of a multivariable prediction model for Individual Prognosis Or Diagnosis (TRIPOD) Statement."
- Abstract (page 2, lines 66-67): "Long-term effects of the TRIPOD statement publication should be evaluated in future studies."
- Introduction (page 4, lines 97-99): "We aimed to assess the difference in completeness of reporting and methodological conduct of published prediction models before and after publication of TRIPOD Statement in high impact general medicine journals."
- Discussion (page 13: lines: 419-421): "Long-term effects of the TRIPOD statement publication should be evaluated in future studies, ideally using the same 2018 TRIPOD Adherence assessment form to allow for comparisons over time."

Furthermore, some authors of our paper have published a pre-TRIPOD baseline measurement of reporting in 146 articles in 37 clinical domains. In a number of years, this will provide an excellent basis for a comprehensive pre-post comparison for the whole medical field to assess the long-term impact of the TRIPOD statement (DOI: 10.1186/s12916-018-1099-2).

Second, the current observational design and data analysis are insufficient to answer the research question stated in the last paragraph of the Introduction part "...to assess the impact of the TRIPOD statement on both...". Even if the authors had observed an improvement, it would be difficult to attribute it to the publication/dissemination of TRIPOD statement, due to potential confounding issues, such as increasing education or awareness. I would suggest that the authors also directly compare pre-TRIPOD studies and non-TRIPOD-citing studies, or studies published on non-TRIPOD-endorsing journals. This paper (PLoS One 2013;8(8):e64733) may also provide a view to address this issue.

We agree with the reviewer that we are not able to assess the actual impact of the TRIPOD statement on the completeness of reporting and methodological conduct of published prediction models. Hence, we have changed the aim of our study to "*evaluate the preliminary difference in completeness of reporting and methodological conduct of published prediction models before and after publication of the TRIPOD statement*". We have accordingly made changes in the title, abstract, and introduction as described in our response to the first question of the reviewer.

We agree with the reviewer that the addition of the comparison between pre-TRIPOD articles and post-TRIPOD articles not citing the statement would be of interest, as it provides information on the

general developments over time. The results of this comparison (pre-TRIPOD: 74% vs post-TRIPOD-non-citing: 76%) were similar to the main pre-post comparison results (74 vs 76%), suggesting a lack of direct impact of the TRIPOD statement on overall reporting levels. We have added this analysis to the methods and results section and have adapted the discussion to these findings.

- Methods (page 6, lines 171-173): “Furthermore to estimate changes over time regardless of the TRIPOD statement, a comparison was made between pre-TRIPOD articles and post-TRIPOD articles not citing the TRIPOD.”
- Results (page 7, lines 197-198): “Results were similar for the comparison between pre-TRIPOD articles and post-TRIPOD articles not citing the statement (76%, absolute difference 2%, 95%CI: -5% to 9%).”
- Discussion (page 13, lines 411-413): “Comparison of articles published before the TRIPOD statement with non-TRIPOD citing articles published after the TRIPOD statement, yielded similar results as the main pre-post comparison, further suggesting a lack of direct impact of the TRIPOD statement on overall reporting levels”

Furthermore, the results might be confounded by other developments in the last couple of years as suggested by the Statistician, such as publication of the PROGRESS BMJ series, and other reports. We have added this information to the discussion section (page 12, lines 366-370): “Furthermore it is not possible to causally attribute the reported changes to the TRIPOD statement, as the results might be confounded by other developments in the last decade, such as publication of multiple series on the conduct of prediction models, publication of other guidelines such as the REMARK guideline for tumor marker prognostic studies, and a general increase in the numbers of published prediction models.[41–43]”

Third, I think the authors need to make a clearer definition on TRIPOD-endorsing journals. I guess the current definition for “endorsing” is a journal that published the original TRIPOD statement in 2015. However, it is possible that a journal did not publish TRIPOD statement but requires all manuscript submission adhering to TRIPOD (usually stated in its author instructions); vice versa. It would be better to differentiate the difference between them. I raised this concern because this study found 18 studies cited TRIPOD but 21 were published on TRIPOD-endorsing journals. This discrepancy looks interesting and may reflect some fundamental issues in exposure definition.

The definition of TRIPOD endorsing journals was indeed that journals published the TRIPOD statement. We have replaced “TRIPOD endorsing journals” with “journals that published the TRIPOD statement” throughout the whole manuscript.

We agree with the reviewer that it would be of interest to compare journals which obligated and did not obligate the authors to complete the TRIPOD checklist. We have therefore added the comparison between articles published in the journals that clearly state that they require adherence to the TRIPOD with journals that did not provide such a statement. We have added this information to the methods and results section.

- Methods (page 6, lines 153-162): “In addition, for all articles published in the post-TRIPOD period, we extracted whether authors cited or referred to the TRIPOD Statement, provided the completed checklist, if the article was published in a journal that published the Statement (The

BMJ, Annals of Internal Medicine, BMC Medicine), or was published in a journal that clearly stated in the author guidelines that they required TRIPOD adherence for submitted work at the time of writing this manuscript (the BMJ, JAMA, and PLOS Medicine). While all included journals (except for the NEJM) encouraged authors to follow the Equator Network guidelines, which includes the TRIPOD checklist, in their author instructions, only the BMJ, JAMA and PLOS Medicine required adherence to the Equator network guidelines and also required to include a filled-out checklist at the time of submission.”

- Methods (page 6, lines 168-173): *“Comparisons were made between articles I) Pre- and Post-TRIPOD, II) post-TRIPOD between articles published in journals that published and did not publish the TRIPOD, III) between articles published in journals that require TRIPOD adherence or not, IV) citing vs. not citing the TRIPOD, and V) providing vs. not providing a completed TRIPOD checklist. Furthermore, to estimate changes over time regardless of the TRIPOD statement, a comparison was made between pre-TRIPOD articles and post-TRIPOD articles not citing the TRIPOD.”*
- Results (page 7, lines 201-205): *“Post-TRIPOD, for articles referring vs. not referring to the statement, published in journals that published vs. did not publish the statement, and published in journals that required adherence to the statement vs. did not require adherence to the statement, no difference in the completeness of reporting was observed (Supplementary Tables 3-5).”*
- We have added Supplementary Table 5: *“TRIPOD reporting scores for articles published after TRIPOD statement in journals that require adherence to the TRIPOD statement and journals that do not require to the adherence statement”*

A limitation is that we do not know if journals required TRIPOD adherence at the time that the included studies were published. We have added this as a limitation to the discussion (page 12, lines 383-385): *“Furthermore, it would have been of interest to compare articles published in journals that between January 2016 and December 2017 obligated authors to complete the TRIPOD checklist, however this information was not available.”*

My other minor concerns are listed below.

1. It seems that the “Recommendations on reporting and methods” in the Discussion provides nothing novel, and I would suggest making this part as concise as possible.

We agree with the reviewer that the reported recommendations are not novel, and have therefore removed almost 40% of the text in this paragraph. We have retained some of these recommendations as we think it is important to provide this information for the non-methodologists and clinicians reading this paper. We now only focus on what we believe are the most important aspects: information on missing data and handling of missing data, reporting of the full model, model development and validation, and calibration and discrimination as performance measures.

2. The authors may need to state in the Discussion that there may be other items for evaluation missed in this study, which is one of the limitations.

While we think we are very comprehensive in our evaluation as we used both the TRIPOD Adherence form for reporting and an extensive self-made list of items for the evaluation of methods, we agree with the reviewer that we might have missed interesting items for evaluation, especially as the field of prediction models is continuously developing. Therefore, we have added the following text to the limitations section (page 12, lines 377-379): *“Although both reporting and methods were*

comprehensively assessed, we might have missed interesting items for evaluation, especially as the field of prediction models is continuously developing.”

3. Page 4 line 105-107, “for articles addressing multiple models but not explicitly recommending a single model, the model with the most predictors was included”. Do these models use the same outcome? Does that mean you would include the full model? It would be better to show an example.

These articles indeed describe multiple models for the same outcome. For example, Hippisley-Cox et. al. (2013) developed three models (A, B and C) to estimate future risk of cardiovascular disease. Model B and C were similar to model A with the addition of extra variables. Model C included the most predictors, and hence we evaluated this model for this publication. An example is provided in the methods section (page 5, lines 123-126): “For instance, Hippisley-Cox (2013) described model A, B and C for the prediction of future risk of cardiovascular disease, with model B being the same as model A with the addition of several predictors and interactions and model C being the same as model B with the addition of one variable. In this case model C was evaluated.”

4. Please unify “TRIPOD adherence form” and “TRIPOD Adherence form” through the manuscript.

We replaced “*TRIPOD adherence form*” with “*TRIPOD Adherence form*” throughout the manuscript.

5. Figure 1, bottom left box “The NEJM: n=n=1” should be “The NEJM: n=1”.

We thank the reviewer for noticing this typographical mistake, which we have fixed.

6. The study assessed whether an item was reported in the main text or supplementary materials, and supplementary table 5 showed items reported in the supplementary materials. I wonder the reporting score represented in figure 2 was based on reporting in the main text only or in main text and supplementary materials both.

For the main analyses and the scores presented in Figure 2 reporting of items in both the main text and supplements were considered. We have added the following text to the methods section to clarify this (page 5, lines 137-138): “For all items and aspects of the checklist it was assessed whether it was reported in the main article or supplementary materials. *The main analyses were based on items reported in either the main text or supplements.*”

7. The authors may consider reporting in Results the percentage of studies that reported any discrimination measures, any classification measure, and any calibration measure.

We agree with the reviewer that it would be interesting for the reader if we report the percentage studies reporting any measure of discrimination, classification and calibration. Therefore, we have added the following information to the results section (page 9, lines 261-262 and lines 265-266):

- *The majority of studies presented measures of calibration (pre-TRIPOD: 66%, post-TRIPOD: 87%) and discrimination (pre-TRIPOD: 91%, post-TRIPOD: 100%, Figure 3 and Supplementary Table 7).*
- *Measures of classification were reported in more than half of the studies (pre-TRIPOD: 69%, post-TRIPOD: 58%)*

Reviewer: 2 (Junfeng Wang)

Recommendation:

Comments:

Zamanipoor Najafabadi et al. compared the adherence to TRIPOD statement before the publication of TRIPOD (pre-TRIPOD) and one year after that (post-TRIPOD). In total 70 articles were assessed (32 pre-TRIPOD vs 38 post-TRIPOD), and no significant improvement in percentage of reported items was found.

From my experience in evaluating reporting quality of prediction models[1] and diagnostic tests[2], this very informative paper represents enormous amount of work done by the authors. So I would like to congratulate them on finalizing all data extraction, statistical analysis and this high quality manuscript.

We would like to thank the reviewer for the effort to review this manuscript and acknowledging the relevance of our work.

I listed my questions below: some are comments and some are for the curiosity of the reviewer and potential audiences.

Abstract

Line 42, page 2, "Quality of methods was assessed with a comprehensive data extraction form based on 43 previous studies, current methodological consensus, and the TRIPOD Exploration and Elaboration document." TRIPOD mainly focused on reporting completeness, although the E&E did include some methodology considerations, it is not the best tool for methodology quality assessment. PROBAST risk of bias assessment tool would be a better choice for this purpose.

We agree with the reviewer that the TRIPOD statement itself primarily focuses on reporting and less on methodology. Therefore, we indeed used multiple existing reviews and current methodological consensus, in addition to the TRIPOD Exploration and Elaboration document, to develop an item list for the assessment of used methodology. At the time of data-extraction the PROBAST was not yet published. In addition, we don't think it would be feasible to score the included articles according to the PROBAST since to do so subject-specific knowledge is required and the included studies span a wide range of clinical subjects. The PROBAST items that we feel are subject to clinical knowledge are 1.2, 3.1, 3.6, 4.1 (reasonable number of participants may differ depending on how rare a disease or outcome is) and 4.6. Furthermore, as the PROBAST only gives suggestions for signaling questions and no scoring rules, we feel that the final risk of bias assessment is slightly too subjective for use in the current study. Although risk of bias and used methods are related, we were primarily interested in the actual used methods. Therefore, we chose to keep our comprehensive item list for the assessment of methods. We have added this information to the discussion section as a limitation (page 12, lines 379-383): "*We also did not assess the risk of bias of the included studies with the PROBAST risk of bias assessment tool, as it would be not feasible to score the included articles according to the PROBAST, since to do so subject-specific knowledge is required and the included studies span a wide range of clinical subjects. Furthermore, as the PROBAST only gives suggestions for signalling questions and no scoring rules, it does not completely fit with the aim to assess the actual used methods of the included studies.*"

Line 53, page 2, “More models were externally validated...” which is not clear enough. Models can be externally validated in other papers as well, I assume this is not the authors meant to say. Maybe better to use “More models were externally validated in the same paper....” here.

We agree with the reviewer that this should be more specified and therefore we have changed the sentence (page 2, lines 58): *“More models were externally validated in the same article”*

Introduction

Line 74, page 3, “Previous systematic reviews on published prediction models have identified poor reporting and many methodological shortcomings in the development and validation of these models.” Systematic reviews of prediction models usually refer to reviews of model performance rather than methodological quality. This sentence should be more specific as “Previous systematic reviews on quality of published prediction models...”.

This sentence should indeed be more specified to methodological systematic review and hence we have changed it (page 4: lines 88-89):

“Previous systematic reviews on the quality of published prediction models have identified poor reporting and many methodological shortcomings in the development and validation of these models.”

Methods

Line 95, page 4, “PubMed search string (Supplementary text 1)”. The reviewer checked the search strategy provided in Supplementary text 1, and suggests to add “nomogram” into the terms. Sometimes, nomogram is used to refer to a prediction model. Although the reviewer does not support this, a lot of researchers still like to use nomogram as the title of a clinical prediction model study, instead of using prediction, prognostic, or diagnostic. The authors may miss some prediction model studies if “nomogram” was not added to the search strategy.

We thank the reviewer for the suggestion and we have updated the literature search with the addition of the search term “nomogram” for the same time period. The addition of this search term did not provide any new articles, therefor the results remain the same.

Line 133, page 5, “In addition, we extracted whether authors cited or referred to the TRIPOD Statement...”. This is a good point, it will be more helpful if information of whether TRIPOD checklist was provided in supplementary material was extracted as well. Implementation of TRIPOD contains several levels: (1) availability/awareness of TRIPOD (pre- vs post-); (2) use of TRIPOD (cited or referred); (3) correct understand of TRIPOD (providing a properly filled TRIPOD checklist. (1) is already achieved by the publication of TRIPOD in 2015, (2) will need promotion of TRIPOD to researchers, and (3) will need more training or workshop provided to researchers. So it is important to check for those authors filled out the checklist (usually this means all items were reported), did they indeed reported completely adherent to TRIPOD, so we can know what actions are needed to improve the reporting quality.

We agree with the reviewer that it might be of interest to compare post-TRIPOD articles that provided a completed TRIPOD checklist as supplement with articles that did not provide it as a supplement. We only found 5 out of 38 post-TIRPOD articles providing the completed TRIPOD checklist. On average these articles reported 80% of items relevant for the manuscript, while 100% of items were marked in the supplement. Due to this very small number, we chose to not perform a statistical test but to report these results descriptively. We have also incorporated this result in to the discussion.

- Results (page 7, lines: 205-206): “Five articles presented the completed TRIPOD checklist in the supplementary material and the overall percentage of reporting for these articles was 80%.”
- Discussion (page 11, lines: 329-331): “Requiring authors to provide and publish the completed checklist might help to improve reporting levels, as we found that the small number of studies that provided the checklist reported more items on average.”

Line 139, page 5, “Reporting levels are presented as percentages...”. The authors also presented the SD together with percentages, please specify the methods for calculating SD. Because the reviewer tried to replicate the calculation of SD by using standard formula but did not get the same number as presented in Supplementary Table 1. And actually, the reviewer did not think it is necessary to add SD in that table.

After reconsideration, we agree with the reviewer that presenting standard deviations is of little added value and therefore we have removed it from Supplementary Table 1.

Previously, the standard deviations were calculated using the standard formula to estimate the standard deviation $\sqrt{\frac{\sum|x-\mu|^2}{n}}$. The quadrated differences between the percentage items reported per article and the average for all articles were summed and then divided by the number of articles. Then the square root was taken. This was done for the overall percentage items reported per article and for the percentage items reported per section (e.g. methods, discussion). Hence, no standard deviation was reported per item.

However, as the percentage of items reported per article were not provided, the reviewer would indeed not have been able to recalculate the SD's.

Results

Line 166, page 6 “However, an improvement for almost half of the individual TRIPOD items (16 items, 42% of items, Supplementary Table 2) “. This sentence is misleading. When the reviewer read it for the first time, it seems a great improvement, but when read it again and looked at Supplementary Table 2, 58% of items were less frequently reported post-TRIPOD. Can we conclude this is an improvement? So the message is too positive, and should be balanced.

We agree with the reviewer that the message was not well balanced. The statistician also named this issue. We would like to refer to our response to the statistician for a detailed report of the changes we have made to accommodate this.

Line 193, page 7, “... almost half of the studies reported to have developed a web application (pre-TRIPOD: 38%, post-TRIPOD: 37%)...”. 38% and 37% are more near to one third than “almost half”. “Almost half” sounds like above 45%...

We have changed the sentence according to the provided feedback of the reviewer (page 8, lines 230-232): *“To improve clinical usability (TRIPOD-item 15b), almost half of the more than one third of studies reported to have developed a web application (pre-TRIPOD: 38%, post-TRIPOD: 37%)”*

Discussion

Line 240, page 9, “a clear improvement in reporting and methodological conduct was found for certain aspects.” As the reviewer already mentioned in Results section, the improvement observed in 16 items is more like by chance. If reporting quality can either increase or decrease, we can always observe improvement in some of the items. So “a clear improvement ... for certain aspects” is just like cherry-picking.

We agree with the reviewer that our main conclusion needs some nuance, as we found a 2% non-significant overall improvement and only improvement for 42% of individual TRIPOD items. However, we do report improvement in almost all of the evaluated methodological aspects: more models were externally validated in the same paper (31 vs 39%), more studies used multiple imputation for handling of missing data (38 vs 50%), less models used statistical cut-offs for variable selection and retaining in multivariable models (67 vs 44%), more studies assessed model performance with model discrimination (91 vs 100%) and calibration (66 vs 87%), and more articles reported the full prediction model (12 vs 41%). We have changed the text accordingly (page: 10, lines 278-284):

“No significant improvement in the overall reporting quality of prediction models published in the seven general medicine journals with the highest impact factor was found in the post-TRIPOD period, according to the TRIPOD Adherence form. However, an improvement in general methodological conduct was found. Notably, more studies described external validation of a model, used multiple imputation methods instead of complete case analysis for handling of missing data, selected and maintained variables in multivariable models based on clinical relevance instead of statistical cut-offs, and assessed both discrimination and calibration measures.”

It will be great if the authors can extensively discuss why reporting quality is still sub-optimal even after the publish of TRIPOD, and what efforts are needed to improve that. That will add value to the current descriptive report.

We agree with the reviewer that it is of added value to discuss possibilities to improve reporting levels for the TRIPOD and other reporting guidelines. We have added the following text to what was already written in the discussion (page 11, lines 329-342). *“These findings pose the question how the introduction and publication of these guidelines can optimally impact the research field. For both the CONSORT and STARD statement, journals endorsing the statement showed a higher level of reporting compared with journals not endorsing these statements. Nevertheless, this was not found for the REMARK guideline, nor in our study for the TRIPOD statement.[34,37,38] Evidence of a relation between citing the statement and reporting level is also limited, as no association between this was found for the STARD nor in our study.[38] Requiring authors to provide and publish the completed checklist might help to improve reporting levels, as we found that the small numbers of studies providing the checklist reported more items on average. Therefore, we do not only recommend journals to ask authors to submit the completed checklist upon submission, but also require authors to publish it as a supplement, and reviewers and editors to control the provided checklist. However, as endorsing, citing and providing the checklist seems to have only a small effect on the reporting quality, we believe it is even more important to train methodologists and clinicians to interpret and use the checklist. This is supported by the results that even studies that provided the completed checklist, still did not report all items of the TRIPOD statement in analysis of reporting. Documents such as the TRIPOD Exploration and Elaboration document facilitate proper interpretation, but we believe that the threshold to use this detailed document might be too high for the unexperienced researcher. Other possibilities to familiarize authors with the checklist should be explored, such as collaborative efforts of educational institutions and the TRIPOD committee to train researchers and clinicians. Online training courses might be of added value to reach a large target group.”*

Tables and Figures

Page 18, Table 1. The heading of Table 1 should be more informative. For example: Analysis, Recommended methods, References.

We have modified the heading of Table 1, so it will better cover the provided information in the table:
“~~Recommendation of preferred~~ Recommended methods and methods analyses for the development and validation of prediction models including supportive references”

Page 19, Table 2. The heading of Table 2 should be consistent with the data in the table: “number (%)”.

To clarify the reported results, we have added to the first row of both columns: “*number (%)*”.

Page 20, Figure 1. Bottom left of the flow chart, “The NEJM: n=n=1“ should be “The NEJM: n=1“

We thank the reviewer for noticing this typographical mistake, which we have fixed.

Page 21, Figure 2. “Ransom split” should be “Random split”; “Fully independent” is not clear, usually “domain validation” is used together with temporal and geographic validations; “Intercept and slope, calibration in-the-lagre, slope”, the first one includes the latter two, do the author mean “Both intercept and slope, only calibration in-the-lagre, only slope”? Why ROC is considered as classification but not part of C-statistic/AUC?

We thank the reviewer for his thorough review of Figure 2.

We have fixed the spelling mistake.

We changed “fully independent” into “non-related cohort”, as we assessed whether the development and validation cohort were directly related to each other or not.

We indeed mean “both intercept and slope”, “only calibration in-the-large” and “only slope”, and have changed it accordingly.

We agree that the ROC should be considered as part of the C-statistics/AUC and have changed the graph accordingly

Supplementary documents

Page 24, Supplementary Table 1. As the reviewer mentioned in Methods section, SD can not be replicated. This also applies to Table 3 and Table 4.

Page 24, Supplementary Table 2. SD is in the heading, but not reported in the table.

We have removed the standard deviations from the table, as the reviewer suggested that it is not necessary to present it.

Page 27, Supplementary Table 6. The heading should be consistent with the data in the table: “number (%)”.

Page 28, Supplementary Table 7. The heading should be consistent with the data in the table: “number (%)”.

Page 29, Supplementary Table 8. The heading should be consistent with the data in the table: “number (%)”.

We have added to the columns “*number (%)*” to clarify the numbers presented in the table

[1] Liu, S., Chen, W., Heus, P., Dong, W., Zhai, W., Cui, Z., & Wang, J. (2019, July). Assessment of reporting quality of prediction model studies in HSCT: Adherence to the tripod statement.

In BONE MARROW TRANSPLANTATION (Vol. 54, pp. 685-685). MACMILLAN BUILDING, 4 CRINAN ST, LONDON N1 9XW, ENGLAND: NATURE PUBLISHING GROUP.

[2] Korevaar, D. A., Wang, J., van Enst, W. A., Leeflang, M. M., Hooft, L., Smidt, N., & Bossuyt, P. M. (2015). Reporting diagnostic accuracy studies: some improvements after 10 years of STARD. *Radiology*, 274(3), 781-789.